# Contribution to the Study of Perioperative Factors Affecting the Restoration of Dog’s Mobility after Femoral Head and Neck Excision: A Clinical Study in 30 Dogs

**DOI:** 10.3390/ani13142295

**Published:** 2023-07-13

**Authors:** Androniki Krystalli, Aikaterini Sideri, George M. Kazakos, Anthi Anatolitou, Nikitas N. Prassinos

**Affiliations:** 1Surgery & Obstetrics Unit, Companion Animal Clinic, School of Veterinary Medicine, Faculty of Health Sciences, Aristotle University, 54627 Thessaloniki, Greece; 2Clinic of Surgery, Faculty of Veterinary Science, School of Health Sciences, University of Thessaly, 43100 Karditsa, Greece; 3Surgery and Anesthesiology—Intensive Care, Companion Animal Clinic, School of Veterinary Medicine, Faculty of Health Sciences, Aristotle University, 54627 Thessaloniki, Greece

**Keywords:** algometer, dog, femoral head and neck excision, multimodal analgesia, ropivacaine

## Abstract

**Simple Summary:**

Femoral head and neck excision is a common and straightforward surgical procedure that provides pain relief for dogs with severe coxofemoral joint disease. Early mobilization of the coxofemoral joint after surgery promotes the development of a fibrous pseudo-articulation with an improved range of motion. In this study, we explored three analgesic protocols (preoperative epidural anesthesia with morphine, intraoperative ropivacaine at the ostectomy site, and postoperative tramadol) either individually or in combination, as part of the standard analgesic protocol for managing 30 dogs undergoing femoral head and neck excision. The aim was to investigate their impact on the time it took for the dogs to bear weight on the limb. Clinical parameters, algometer measurements, and various scale scores were used to evaluate the outcomes. The study concluded that multimodal analgesia, incorporating the aforementioned analgesic techniques, resulted in faster weight bearing for dogs with femoral head and neck excision.

**Abstract:**

This study aimed to compare postoperative analgesia and the time of limb weight bearing induced by the intraoperative administration of a local anesthetic at the site of the femoral head and neck excision (FHNE) in dogs, with and without the administration of preoperative epidural anesthesia. Additionally, the impact of postoperative opioid drug administration on weight-bearing time was examined. This randomized, blinded, prospective clinical study included 30 client-owned dogs. The dogs were randomly divided into three groups (A, B, C), each further divided into two subgroups (A1, A2, B1, B2, C1, C2). Group A received epidural anesthesia and ropivacaine at the ostectomy site, Group B received only ropivacaine, and Group C served as the control group. Subgroup 1 received a non-steroidal anti-inflammatory drug postoperatively, while Subgroup 2 had tramadol added to their regimen. Pain assessment was conducted using the University of Melbourne Pain Scale (UMPS) and an algometer. The study concluded that multimodal analgesia, utilizing all the aforementioned analgesic techniques, resulted in faster limb weight bearing for dogs undergoing FHNE.

## 1. Introduction

Femoral head and neck excision (FHNE) is a surgical salvage procedure [1,2,3,4,5]. It involves the removal of the entire femoral head and a portion of the femoral neck through an oblique osteotomy, starting medially at the greater trochanter and ending proximally at the lesser trochanter [6]. The primary goal of this procedure is to eliminate bone-to-bone contact between the femur and acetabulum, thereby restoring proper and pain-free coxofemoral joint function. By eliminating abnormal crepitus between the articular surfaces of the “coxofemoral joint” [7,8], the joint is replaced by a functional pseudarthrosis, consisting of dense fibrous connective tissue lined by a synovial membrane [9,10]. The acetabulum and proximal femur continue to remodel over several years following the surgery [10].

FHNE was initially described by Girdlestone in humans as a means to alleviate pain caused by coxofemoral joint tuberculosis [11] and coxofemoral septic arthritis [12]. Subsequently, veterinary surgeons modified and adapted this technique for use in dogs and cats [13,14]. The outcomes of these adaptations were deemed advantageous for managing diverse coxofemoral disorders [15].

FHNE is indicated for various conditions, including hip dysplasia, avascular necrosis of the femoral head (Legg Calvé Perthe’s disease), fractures of the femoral head and/or neck, acetabular and/or pelvic fractures (often associated with significant soft tissue trauma), non-reducible or chronic coxofemoral luxation, severe coxofemoral osteoarthritis with significant clinical impact, and failed total coxofemoral arthroplasty [6,16].

Several perioperative factors influence the outcome of FHNE. Preoperative factors encompass the patient’s body weight, age, and the chronicity of the condition that necessitated FHNE. Intraoperatively, the surgical technique employed plays a crucial role [4,17]. Employing an atraumatic surgical approach, performing a proper and precise neck ostectomy, and creating a smooth resection surface facilitate painless bone contact, thereby expediting the return to weight bearing on the limb [5,18]. Postoperatively, the use of analgesics, controlled physical activity, and physiotherapy are positive factors contributing to the outcome following FHNE [4,6,19]. Prompt resumption of function after surgery is necessary to form functional pseudoarthrosis. Delayed return to mobility leads to restricted joint motion and unfavorable long-term outcomes [8].

Multimodal analgesia, involving the early administration of appropriate analgesics using multiple drug classes and/or techniques, is recognized as a strategy to prevent or minimize pain, such as in FHNE [20]. Effective pain control postoperatively is a crucial objective of anesthesia [21]. Preemptive analgesia involves initiating pharmacological analgesia before the painful stimulus, inhibiting nociceptive mechanisms before they are activated. The duration of the block should encompass the entire surgical and postoperative periods [22]. Preemptive analgesia focuses on the timing of drug administration rather than the specific drug choice. Administering a drug preoperatively yields more significant analgesic effects compared to postoperative administration.

Additionally, preventive analgesia suggests administering drugs solely during the surgical procedure and not before its commencement. Preventive analgesia aims to reduce postoperative pain or analgesic consumption, obviating the need for additional drugs [23]. Moreover, if drugs are administered before or at the end of the surgery, the dosage or administration method may need to be adjusted [24,25]. While some studies have not demonstrated the beneficial effects of preemptive analgesia in postoperative pain control, others have shown improved pain management and decreased analgesic consumption during the postoperative period. This was one of the motivations for the present study. Achieving a satisfactory level of postoperative pain control necessitates maintaining patient homeostasis throughout all phases of the perioperative period [21]. Furthermore, multimodal analgesia involves utilizing multiple analgesic medications to target various points along the pain pathway.

Local anesthetics have been utilized in clinical practice for over a century, as they induce a reversible blockade of nerve conduction in a specific area, resulting in loss of sensation [26]. Epidural analgesia can be employed for surgeries performed caudal of the diaphragm level. Its use is particularly valuable for highly painful procedures like coxofemoral joint surgeries, owing to its excellent postoperative analgesic effects [27,28]. Epidural administration of opioids offers prolonged analgesia with fewer adverse effects than systemic administration. Combining opioids with local anesthetics in epidural anesthesia is a widely adopted approach to achieve regional anesthesia. By doing so, opioids exert their analgesic properties through various mechanisms, both within the spinal cord and independently of their action in the spinal cord [29].

In veterinary medicine, morphine is the opioid most frequently employed in epidural anesthesia, primarily due to its high potency and prolonged duration of action. It is administered either as a standalone medication or in combination with local anesthetics [30,31,32].

Alternatively, various techniques are being developed to provide local anesthesia/analgesia around the intervention site. In humans, a method called hematoma block involves injecting a local anesthetic directly into the hematoma of a fracture, facilitating closed reduction [33]. This technique has also been applied to fracture osteosynthesis to ensure postoperative analgesia in both humans [34] and dogs [35]. Furthermore, when combined with procedural sedation and analgesia, hematoma block can yield satisfactory pain relief and reduce the need for additional analgesic medication post-procedure [36]. Complications associated with hematoma block are rare in people but can include infection, osteomyelitis, lidocaine toxicity, cardiotoxicity, seizures, and an increased risk of compartment syndrome [37,38,39,40]. These complications can manifest as symptoms such as rhinitis, dizziness, numbness around the mouth or tongue, changes in mental status, confusion, cardiac arrhythmias, and hypotension [33].

Similar analgesic techniques have been proposed for canine ovariohysterectomy (OHE). Carpenter et al. (2004) [41] used intraperitoneal and incisional bupivacaine to reduce postoperative pain following traditional OHE. In a subsequent study, Kim et al. (2012) [42] investigated the effect of intraperitoneal bupivacaine on postoperative pain behavior and biochemical stress in dogs undergoing laparoscopic OHE, reporting significant and encouraging results. Furthermore, Shilo-Benjamini (2018) [43] documented positive outcomes after applying bupivacaine splash into the empty orbit following globe removal.

Among tools that used in various studies to assess pain are the UMPS [44] and the algometer. The latter is an electronic device used for the quantitative measurement of the mechanical pain threshold based on Von Frey filaments [45]. It consists of the main stem, handle, memory recorder, and nozzles (tips). A metallic probe is screwed into the main stem and, with pressure, activates the actuator. Finally, a plastic polypropylene tip with length 3 cm and 0.78 mm^2^ disk surface area is adjusted to the probe (Figure 1). Gradual pressure is applied at a specific point to produce a non-traumatic painful mechanical stimulus. Specifically, the pressure is applied to the skin through the nozzle, and the recording device numerically displays and stores the maximum reading in grams. The measurement uses a highly accurate sensor of with a maximum capacity of 5000 g. This ensures the precise capture and display of the maximum force, even for short and low force peaks, enabling the testing of a wide range of the animal’s pain sensitivity.

To understand the methodology and findings of the study, it is essential to define the two parameters employed. The time of initial weight bearing (TIWB) of the limb refers to the moment during the postoperative period of FHNE when the animal first began using the limb, albeit with varying degrees of lameness (partial weight bearing). Conversely, the time of final weight bearing (TFWB) of the limb signifies the moment, within the postoperative period of FHNE (maximum follow-up period: 1 year), when the limb exhibited full weight bearing or displayed the least amount of lameness possible (partial weight bearing), which remained unchanged until the study’s completion.

The research hypotheses of the study were as follows: (a) The intraoperative administration of a local anesthetic at the ostectomy site leads to a shorter TIWB and subsequent TFWB during the postoperative period of FHNE. Furthermore, the additional use of preoperative epidural anesthesia further enhances these timeframes. (b) Dogs that receive an opioid drug along with standard non-steroidal anti-inflammatory drug (NSAID) therapy postoperatively exhibit significantly shorter TFWB following FHNE.

## 2. Materials and Methods

This study is a blinded, randomized, prospective clinical trial that involved 30 client-owned dogs presented to the Surgery and Obstetrics Unit at the Companion Animal Clinic, Department of Veterinary Medicine, Aristotle University of Thessaloniki, Greece, from 2018 to 2022. The dogs were suffering from coxofemoral problems and were treated with FHNE. The animal study protocol was approved by the Research and Ethics Committee of the Department of Veterinary Medicine, Aristotle University of Thessaloniki, Greece (protocol no. 567/13-3-2018). Written consent was obtained from the owners of each participating animal.

### 2.1. Animals

A comprehensive clinical, orthopedic, and neurological examination was conducted on all animals, along with radiographic imaging and basic hematological and biochemical blood testing.

The following criteria were used to exclude animals from the research protocol: (i) age younger than 4 months due to the long NSAID intake period; (ii) abnormalities in basic hematological and biochemical parameters; (iii) lameness caused by factors other than the local problem in the coxofemoral joint (e.g., neurological disorder, osteoarthritis of the stifle or tarsus); (iv) lameness affecting other limbs; (v) incomplete elimination of any previously administered anti-inflammatory/analgesic drug (based on the drug’s pharmacokinetics); (vi) inability to follow-up with postoperative evaluations for at least 1 year; and (vii) owners unwilling to comply with the required re-examinations program.

The animals were randomly divided into three groups, each consisting of ten dogs (Groups A, B, and C). Within each group, there were two subgroups, each containing five dogs (Subgroups A1, A2, B1, B2, C1, and C2) (Table 1). Group C was designated as the control group. The following information was recorded upon the initial presentation of all animals: the underlying condition necessitating FHNE, the duration of the condition, the age of the animal, and its body weight.

### 2.2. Clinical Examinations

During each examination (initial presentation and re-examinations), the following procedures were conducted: a comprehensive physical, neurological, and orthopedic examination was performed on all dogs; visual gait assessment was conducted by observing and video recording the dogs while they were standing, walking, and running (Table 2) always by the first author (AK). Running was excluded from the first two re-examinations.

The first three re-examinations were conducted on the 15th, 30th, and 60th days, followed by monthly intervals until the TFWB of the limb. The maximum duration of the follow-up period was 12 months.
Ventrodorsal and, if necessary, lateral radiographs of the coxofemoral joints were taken during the initial and final examinations while the dogs were under general anesthesia.During the initial examination and the first three re-examinations, when long-term administration of NSAIDs was required, all dogs underwent hematological and biochemical examinations, including a complete blood count and the assessment of parameters such as alkaline phosphatase (ALP), alanine aminotransferase (ALT), albumin, creatinine, and glucose levels. If NSAID administration continued for an extended period, liver and kidney function tests were conducted every 3 months.

### 2.3. Pre-Anesthetic Period

Each experiment involved a pre-anesthetic fasting period of 6–12 h, depending on the type of last meal (canned or dry food), and access to water was allowed up to 2 h before the operation. The pre-anesthetic examination focused on the following in addition to routine procedures:Heart and respiratory rates;Pain measurement using Von Frey filaments (Vetalgo Algometer, Bioseb, BP 32025, F-13845 Vitrolles Cedex, France). Measurements were taken at: (i) the ostectomy site (OS) (immediately in front of the greater trochanter), (ii) a “healthy” area of the affected limb near the ostectomy site (NOS) (distal to the greater trochanter toward the femur), and (iii) the corresponding ostectomy area of the contralateral healthy limb (CHL) for comparison with postoperative findings. Three measurements were taken in each area, and the average was recorded.

In this study, all measurements with the algometer were conducted with minimal animal restraint. For each measurement, the pressure applied by the nozzle on the skin was immediately stopped upon the first reaction of the animal to the painful stimulus. The animal’s response, such as limb withdrawal and/or vocal manifestations, was considered a reaction to the painful stimulus.

Table 1 summarizes the anesthetics and analgesic drugs administered to the animals. The pre-anesthetic regimen was consistent across all groups and involved the intramuscular administration of an α_2_-adrenoreceptor agonist (150 µg m^2^ bodyweight) (Dexdomitor; Elanco, Athens, Greece) and subcutaneous administration of robenacoxib (2 mg kg^−1^ bodyweight) (Onsior; Novartis Animal Health, London, UK). After intravenous catheter insertion, tramadol (3 mg kg^−1^ bodyweight) (Tramal; Grunenthal, Aachen, Germany), and a 2nd-generation cephalosporin (30 mg kg^−1^ bodyweight) (Cefur; Fresenius Kabi Hellas, Athens, Greece) were administered intravenously. The lumbar region of all animals was prepared for epidural anesthesia, but it was only performed in Group A using morphine (0.1 mg kg^−1^ bodyweight) (Morphine sulfate 10 mg mL^−1^; State Drug Monopoly, Famar S.A., Athens, Greece). Propofol (1 mg kg^−1^ bodyweight) (Propofol MCT/LCT; Fresenius Kabi Hellas) was used bolus intravenously, followed by additional doses (0.5 mg kg^−1^ bodyweight) to effect tracheal intubation. Anesthesia was maintained using isoflurane (2%) (Isoflurane-Vet; Merial, Milano, Italy) in oxygen at a flow rate of 2 L min^−1^ through a suitable circuit.

### 2.4. Surgical Technique

To perform FHNE, the animal was positioned in lateral recumbency with the affected limb positioned upward. Clipping and aseptic preparation of the surgical field focused on the greater trochanter and extended to the middle of the tibia on the affected limb. The preferred approach to the coxofemoral joint was a craniolateral approach [4,5,46]. The joint capsule was incised, and the femoral head was detached from the intact round ligament, if present, using a bifurcated syndesmotome. Hohmann retractors were utilized to support and adequately expose the femoral head and neck. For the ostectomy, an osteotome and hammer were employed. The excision line started proximally from the medial aspect of the greater trochanter and ended distally at the proximal aspect of the lesser trochanter [6]. The lesser trochanter was preserved in all cases. The surface of the femoral neck was smoothed using an orthopedic rasp. The absence of crepitus during the passive range of flexion and extension of the coxofemoral joint indicated sufficient smoothing of the OS [5]. The ostectomy was performed by the first author (AK) in all animals. Intraoperatively, immediately after the ostectomy, the anesthetist injected ropivacaine (1 mg kg^−1^ bodyweight, 0.13 mL kg^−1^ final volume) (Ropivacaine Kabi 0.75% solution; Fresenius Kabi, Halden, Norway) into the surgical field in Groups A and B, or normal saline (0.13 mL kg^−1^ bodyweight) (Sodium chloride 0.9%; Vioser, Trikala, Greece) in Group C, and allowed it to take effect for 5 min before tissue closure. The procedure was “blinded” to the first author (AK), who also conducted the postoperative pain assessments. Using routine techniques, the surgical site was closed in layers (muscles, subcutaneous tissue, and skin) [46]. Intradermal skin sutures were applied to all animals, eliminating the need for Elizabethan collars.

### 2.5. Postoperative Period

“Pain measurements” (as defined below) were conducted immediately after transferring the animal to the recovery room, approximately 15–20 min (not exceeding 30 min) after the conclusion of the operation and anesthesia (t_0_). Additional measurements were taken at 1 (t_1_), 2 (t_2_), 4 (t_3_), 6 (t_4_), 20 (t_5_), and 24 h (t_6_) postoperatively. Each “pain measurement” included the following procedures:The UMPS [44] assessed and recorded the corresponding pain level. The UMPS score ranged from 0 (indicating no pain) to 27 (representing the most severe pain). If any measurement yielded an UMPS score ≥15, rescue analgesia was administered, which included morphine (0.3 mg kg^−1^ bodyweight) (Morphine sulfate 10 mg mL^−1^; State Drug Monopoly, Famar S.A., Athens, Greece) intramuscularly and fentanyl (3 μg kg^−1^) (Fentanyl; Janssen Pharmaceutica NV, Beerse, Belgium) intravenously.The threshold was measured using Von Frey filaments in the three previously reported areas.The degree of sedation was evaluated using numerical scoring as follows: zero indicated the animal was fully alert, one described a “dizzy” animal that reacted easily to acoustic and visual stimuli, two denoted a “sleeping” animal that showed minimal responsiveness to acoustic and visual stimuli, and three represented a “sleeping” animal that did not react to acoustic and visual stimuli [35].

The animals were hospitalized for 24 h before being released. At the 20 h mark postoperative, the initial physical therapy exercises were conducted. The postoperative treatment regimen involved restricting the animal’s activity for a month. The animal was allowed to walk slowly on a short leash, but running and jumping were prohibited. Encouraging limb weight bearing was prioritized starting from the first day after the operation. To achieve this, simple physiotherapy exercises involving flexion–extension movements of the hip, stifle, and tarsus were performed 20–30 times, 2–3 times a day, beginning on the first postoperative day. However, since full weight bearing of the limb had not been achieved after the second re-examination (at 1 month), the physiotherapy exercises were expanded to include swimming, back walking with the rear limbs, climbing stairs, and so on. Each owner was meticulously and patiently instructed in the performance of the physiotherapy exercises.

Subgroups A1, B1, and C1 received the NSAID robenacoxib (2 mg kg^−1^ bodyweight) (Onsior; Novartis Animal Health, London, UK) orally once a day until the limb achieved TFWB following FHNE. Animal owners were informed about the potential adverse effects of NSAIDs, and if such effects occurred, administration would be halted, and the animal would be withdrawn from the study. Subgroups A2, B2, and C2 were administered the NSAID along with the opioid drug tramadol (3 mg kg^−1^ bodyweight) (Tramal; Grunenthal GmbH, Aachen, Germany) orally every 8 h for a duration of 15 days.

Postoperative re-examinations were conducted according to the previously described protocol.

### 2.6. Statistical Analysis

A power analysis was conducted to ensure an ethically acceptable study while minimizing the number of animals required to achieve the scientific objectives. We established from a prior study that a cohort of 30 dogs would yield a minimum of 80% power in detecting differences at a significance level of *p* < 0.05 [35].

To investigate correlations between pairs of quantitative variables, we calculated and evaluated either Spearman’s rank correlation coefficient (*rho*) or Pearson’s linear correlation coefficient (*r*) as appropriate. To examine the relationship between qualitative (or categorical) variables, we employed the chi-square test of independence (χ^2^ test). The *p*-value for the observed significance level of the χ^2^ test was determined using the Monte Carlo simulation method, with 10,000 resampling iterations. To compare the three groups of animals in terms of quantitative variables relative to central tendency.

To compare the central tendency among the three groups of animals, along with the two subgroups, concerning the TIWB and TFWB of the operated limb, we employed a two-factor ANOVA method. The first factor, “Group”, consisted of three levels, while the second factor, “Subgroup”, consisted of two levels. Additionally, we considered a within-subject factor, “within”, with two repeated measurements (initial vs. final) applied to the animals.

To compare the central tendency among the three groups of animals, both preoperatively and at time points t_0_–t_6_, regarding mechanical pain threshold measurements at the OS, measurements in a “healthy” area NOS, measurements in the CHL, and pain intensity measurements on the UMPS scale (excluding the preoperative measurement), we employed a two-way ANOVA method. This analysis involved one factor, “Group”, with three levels, as well as a within-subject factor considering repeated measures, “Time”, with eight levels (or seven for pain intensity measurements on the UMPS scale).

In all cases, multiple comparisons of mean values were conducted using the least significant difference (LSD) test. The ANOVA method was employed within the framework of general linear models to ensure accurate estimation of standard errors for the differences between means being compared, ensuring the validity and reliability of the comparison results. The models’ assumptions were as follows: (a) normality of the residuals’ distribution, (b) homoscedasticity of the residuals, and (c) sphericity or homogeneity of variances in the differences of measurements between pairs of time points among the three groups. Assumption (c) specifically applies to the ANOVA method with repeated measures over time. No significant deviations from the assumptions mentioned above were detected. Statistical analyses were performed using IBM SPSS Statistics v.24 software (IBM Corp., Armonk, NY, USA), with the Exact Tests module installed for implementing the Monte Carlo simulation method. In all statistical hypothesis testing procedures, a preset significance level of *p* ≤ 0.05 was used.

## 3. Results

### 3.1. Perioperative Factors

The clinical study involved 30 client-owned dogs, comprising 18 males and 12 females. Among them, only four dogs had been neutered. Most dogs were of mixed breeds (Mongrel), accounting for 19 dogs or 63.3% of the total (Table 3).

The age of the animals ranged from 5 to 120 months, with a mean of 32.46 and a median of 13.5. Specifically, in Group A, the dogs were aged from 5 to 120 months, with a mean of 29.2 and a median of 9.5. In Group B, the age range was from 5 to 96 months, with a mean of 31.3 and a median of 14.5. In Group C, the dogs’ ages ranged from 5 to 108 months, with a mean of 36.9 and a median of 27.

The body weight of the dogs in the study ranged from 2.5 to 34 kg, with a mean of 18.04 and a median of 18.25. In Group A, the dogs weighed between 2.8 and 34 kg, with a mean of 18.3 and a median of 19.7. In Group B, the weight range was from 2.5 to 31 kg, with a mean of 19 and a median of 20.3. Lastly, in Group C, the dogs’ weights ranged from 7.8 to 28 kg, with a mean of 17 and a median of 12.8.

The diseases responsible for FHNE included coxofemoral luxation (*n* = 10), femoral head and/or neck fracture (*n* = 10), hip osteoarthritis (*n* = 5), hip dysplasia (*n* = 4), and aseptic necrosis of the femoral head (*n* = 1). The duration of the disease leading to FHNE ranged from 5 to 420 days, with a mean of 61.1 and a median of 28.

The preoperative lameness of the affected limb was assessed on a six-point scale (Table 2), ranging from 2 to 5, with a mean of 3.23 and a median of 3. After the surgery, all dogs initially exhibited partial weight bearing (mean 2.7, median 3), while at the final evaluation, 86.7% of the dogs achieved full weight bearing, and 13.3% continued with partial weight bearing (mean 2, median 2).

### 3.2. Comparison of the Groups Based on the Animal Data (Sex, Breed, Age, Body Weight, Gonadectomy) and the Disease That Led to the FHNE

Based on the χ^2^ test, there was no statistically significant difference observed between the three groups (A, B, C) in terms of sex (χ^2^ = 2.500, df = 2, *p* = 0.456), gonadectomy (χ^2^ = 0.577, df = 2, *p* = 1.000), breed (χ^2^ = 16.684, df = 16, *p* = 0.388), and the dog’s disease leading to FHNE (χ^2^ = 10.500, df = 8, *p* = 0.239). Additionally, the results of the analysis of variance indicated no statistically significant difference among the groups in terms of the dog’s age (*p* = 0.890) and body weight (*p* = 0.882).

### 3.3. Time of Weight Bearing of the Limb Subjected to FHNE

#### 3.3.1. Time of Initial Limb’s Weight Bearing

The time interval with toe-touching weight bearing for the limb undergoing FHNE ranged from 1 to 25 days, and the mean duration for each subgroup is provided in Table 4. The comparison of subgroups based on TIWB is presented in Table 5.

Regarding the impact of intraoperative administration of ropivacaine on TIWB of the limb, it appears to lead to a shortened duration, as there is a statistically significant difference between Subgroups B1 and C1 (*p* = 0.003). However, the effect of epidural anesthesia in animals receiving intraoperative ropivacaine did not influence TIWB, as no statistically significant difference was found between Subgroups A1 and B1 (*p* = 0.591). Similarly, the postoperative administration of tramadol in animals treated with ropivacaine did not result in a shortened TIWB, as there was no statistically significant difference between Subgroups B1 and B2 (*p* = 0.878). Lastly, the postoperative administration of tramadol in animals that received preoperative epidural anesthesia and intraoperative ropivacaine did not affect TIWB, as there was no statistically significant difference between Subgroups A1 and A2 (*p* = 0.153).

The above findings are supported by comparing Subgroups A1 and C1, revealing a statistically significant difference between them (*p* = 0.001). This suggests that the combination of epidural anesthesia with intraoperative ropivacaine administration at the OS reduced TIWB. Additionally, a statistically significant difference was observed between Groups A2 and C1 (*p* < 0.001) when the aforementioned analgesic techniques were combined with postoperative tramadol administration, resulting in a shorter duration of weight bearing on the operated limb.

The postoperative administration of tramadol reduced TIWB on the operated limb, as indicated by the statistically significant difference observed between Subgroups C1 and C2 (*p* = 0.008). However, the impact of intraoperative ropivacaine administration compared to postoperative tramadol administration on TIWB of the operated limb did not reach statistical significance, as seen in the comparison between Subgroups B1 and C2 (*p* = 0.645).

#### 3.3.2. Time of Final Limb’s Weight Bearing

The range of final weight bearing of the limb following FHNE was from 30 to 90 days, with the mean duration for each subgroup listed in Table 6. The comparison of subgroups in terms of TFWB is presented in Table 7. Throughout this period, the animals received NSAIDs and underwent physiotherapy.

The administration of tramadol during the postoperative period did not have a significant effect on the TFWB of the operated limb, as indicated by the non-statistically significant difference observed between Subgroups C1 and C2 (*p* = 0.501). Additionally, the inclusion of intraoperative ropivacaine administration as an additional component to the analgesic regimen did not impact TFWB, as demonstrated by the non-statistically significant difference between Groups B2 and C1 (*p* = 0.068).

The intraoperative administration of ropivacaine did not have a significant effect on TFWB, as there was no statistically significant difference observed between Subgroups B1 and C1 (*p* = 0.241). However, when comparing Subgroups A1 and C1, a statistically significant difference was found (*p* = 0.020), suggesting that the combination of epidural anesthesia with intraoperative ropivacaine administration at the OS reduces the TFWB of the operated limb. As anticipated, a statistically significant difference was also observed between Groups A2 and C1 (*p* < 0.001) when the aforementioned analgesic techniques were combined with postoperative tramadol administration, resulting in a shorter duration of TFWB on the operated limb.

#### 3.3.3. Correlation between TIWB and TFWB of the Operated Limb

The statistical analysis reveals a robust positive and statistically significant correlation between TIWB and TFWB of the operated limb (r = 0.639, *p* < 0.001). Additionally, there is a noteworthy and statistically significant difference (*p* < 0.001) in effect on TFWB when comparing the combination of intraoperative ropivacaine administration, preoperative epidural anesthesia, and postoperative tramadol administration to the individual effects of ropivacaine (*p* < 0.001) or tramadol on TIWB of the operated limb. Figure 2 shows the correlation between TIWB and TFWB of the limb for all subgroups in the present study.

### 3.4. Pain

In the preoperative measurements of the mechanical pain threshold, no statistically significant differences were observed among the three groups (Table 8). Additionally, none of the dogs included in this study necessitated rescue analgesia due to a high UMPS score, which led to their exclusion from subsequent pain measurements.

Statistical analysis was conducted on the postoperative measurements of the mechanical pain threshold and UMPS pain intensity using two approaches. Firstly, the measurements were standardized concerning time to examine group differences overall, independent of the individual impact of time on the variables of interest. Secondly, a multivariate analysis of repeated measures was performed to account for time variation. For the former approach, the area under the curve was computed for each dog using the trapezoidal rule [47] and then standardized in terms of time, resulting in the time-standardized area under the curve (AUCst). Consequently, derived variables such as AUCstOS (ostectomy site), AUCstNOS (healthy area near the ostectomy site), AUCstCHL (contralateral limb), and AUCstUMPS (UMPS scale) were obtained.

#### 3.4.1. Time-Standardized Area under the Curve

The statistical analysis of variance revealed that there were no statistically significant differences between the groups for the variables AUCstOS (*p* = 0.461) and AUCstCHL (*p* = 0.622). However, significant differences were observed for the variables AUCstNOS and AUCstUMPS. Specifically, for the variable AUCstNOS, a statistically significant difference was found between Groups A and C (*p* = 0.009). As for the variable AUCstUMPS, the mean differences among groups were as follows: Group A had a lower mean than Group B, and the difference was statistically significant (*p* = 0.018); Group A had a lower mean than Group C, and the difference was highly statistically significant (*p* < 0.001); and there was no significant difference between Group B and Group C (*p* = 0.104).

#### 3.4.2. Multivariate Analysis of Repeated Measures

##### Measurements of Mechanical Pain Threshold at the OS

The analysis of the results demonstrated statistically significant differences in the mechanical pain threshold between the animals in Groups A and C at multiple time points. Specifically, significant differences were observed at t_0_ (*p* = 0.004), t_1_ (*p* = 0.0010), t_2_ (*p* = 0.022), and t_4_ (*p* = 0.022), with Group C exhibiting a lower pain threshold compared to Group A. On the other hand, between Groups A and B, a statistically significant difference in the mechanical pain threshold was found only at the time point t_2_ (*p* = 0.045) (Figure 3).

##### Measurements of Mechanical Pain Threshold in a “Healthy” Area NOS

The analysis showed a statistically significant difference in the mechanical pain threshold between the animals in Groups A and C at multiple time points. Specifically, significant differences were found at t_0_ (*p* = 0.009), t_1_ (*p* = 0.011), t_2_ (*p* = 0.011), t_3_ (*p* = 0.043), t_4_ (*p* = 0.027), t_5_ (*p* = 0.049), and t_6_ (*p* = 0.035), with Group C exhibiting a lower pain threshold compared to Group A (Figure 4).

##### Mechanical Pain Threshold Measurements in the CHL

No statistically significant difference was found among the three groups in measuring the CHL (*p* = 0.804) (Figure 5).

##### Measurements of Pain Intensity on the UMPS

The analysis demonstrated a statistically significant difference in pain intensity between Groups A and C (*p* = 0.002), with Group C having a higher average than Group A. Figure 6 illustrates that pain intensity increased across all groups between time points **t_0_** and **t_1_**. The animals in Group C exhibited higher values compared to the other groups. However, none of the animals had a UMPS score exceeding 15 points, indicating that rescue analgesia was not required for any of them.

## 4. Discussion

FHNE is utilized as a final option to alleviate pain caused by an inappropriate hip joint, leading to a significant enhancement in the quality of life for dogs and cats [48]. Specifically, it promotes a suitable and pain-free hip function eliminating the abnormal crepitus between the joint surfaces [7,8,9,10,49].

FHNE is recommended when the hip joint’s anatomical and/or functional integrity cannot be restored [4]. In our study, the indications for FHNE aligned with those proposed by various researchers in terms of the conditions it addresses, such as aseptic necrosis of the femoral head, hip luxation, hip dysplasia, acetabular fractures, femoral head and/or neck fractures, and severe osteoarthritis [5,6,50]. The incidence of these conditions also corresponded to previous findings [16,51,52]. Consequently, the most common indications observed were hip luxation and fractures of the femoral head and/or neck, while hip dysplasia, osteoarthritis, and aseptic necrosis of the femoral head were less frequent. In our cases, the absence of acetabular fractures may be purely coincidental, potentially attributed to the limited number of dogs included in the study.

Based on six decades of literature documenting the use of FHNE, there is a notable success rate and high satisfaction among pet owners. For many individuals, this procedure has served as a salvage operation and a treatment that effectively alleviated their dog’s pain [16,17,53]. Our own experience aligns with these positive outcomes, even though the recovery time for many animals can be unpredictable. As demonstrated by the results, the majority of cases yielded favorable outcomes. However, we recognize the potential for further improvement, prompting us to investigate in this regard.

Although FHNE has been described and widely used by veterinarians for many years, there is still considerable room for improvement in terms of the speed of animal gait recovery as ongoing efforts are made to enhance the technique. Over the past few decades, various techniques have been proposed for this purpose. One such technique involves placing fat or a muscle flap between the ostectomy site and the pelvis. Muscle transposition, particularly using the biceps femoris muscle [54,55], the cranial part of the deep gluteal muscle [3,16,56], or a section of the rectus femoris muscle [57], has been described. Additionally, suturing of the joint capsule after completing the ostectomy has been suggested [46,52]. Another technique involves resecting the lesser trochanter to reduce bony contact between the acetabulum and femur and minimize the risk of sciatic nerve entrapment, thus reducing postoperative lameness [58]. However, the adhesion of the iliopsoas to the lesser trochanter provides stability to the resected femur [59], reducing bone shortening and lameness [60]. Interestingly, none of the aforementioned techniques significantly differ from the classic FHNE technique.

In our study, the modifications we considered for FHNE were focused on the analgesic aspect rather than the surgical procedure itself. The primary aim of this study was to investigate the impact of pain on the postoperative weight-bearing capacity of the operated limb. We hypothesized that effective pain control during the early postoperative period would expedite the achievement of partial and total weight bearing. Additionally, existing literature supports the notion that early restoration of limb function is crucial for developing functional pseudarthrosis [8]. With this in mind, we selected potent analgesic medications and explored different routes and timing of administration, anticipating that they would contribute to improved outcomes for FHNE.

In our study, our objective was to demonstrate the progressive reduction of TIWB and, consequently, TFWB in the limb undergoing FHNE through the implementation of three analgesic interventions: (a) intraoperative injection of local anesthetic on the OS, (b) adjunctive application of epidural anesthesia, and (c) postoperative administration of an opioid in combination with a traditional NSAID. To achieve this, we structured our study by dividing the groups of dogs into different stages, each receiving a specific analgesic regimen (refer to Table 1). This approach aimed to induce analgesia at multiple points along the pain pathway, known as multimodal analgesia, to enhance the outcomes of FHNE. It should be noted that the effects of multimodal analgesia in acute pain have not been extensively studied [61]. However, based on our findings, this type of analgesia appears to positively impact acute pain resulting from a procedure such as FHNE.

In particular, one-third of the animals undergoing FHNE received preoperative epidural anesthesia with the administration of morphine. Morphine was chosen as it is highly potent and has a prolonged duration of action when administered in epidural space. When administered to dogs at a dose of 0.1 mg kg^−1^, it exhibits an onset time of 20–60 min and a duration of action lasting 16–24 h [62,63]. Additionally, morphine has been associated with reduced concentrations of inhaled anesthetics in dogs [64,65]. While there are limited reports on the use of epidural anesthesia for FHNE in dogs [31,66,67], none of them have explored its relationship with the postoperative period. Therefore, our study serves as a foundation for investigating the effects of epidural anesthesia (morphine ± local anesthetic) on the limb’s TFWB.

During the intraoperative phase, local anesthetic was injected into the femoral OS in two-thirds of the animals included in our study. To the best of our knowledge, this is the first description of such an injection in dogs. A similar approach involving local anesthetic injection has been documented for the osteosynthesis of long bone fractures in dogs [35]. In that context, direct injection of bupivacaine into the fracture site enhanced postoperative analgesia. In our research, we opted for ropivacaine due to its rapid onset (approximately 5–10 min) and sufficiently long duration of action (4–6 h) for diffusion techniques [68]. Compared to lidocaine, ropivacaine has a prolonged duration of action [69], which we believed would provide superior postoperative analgesia. Additionally, bupivacaine carries a higher risk of cardiotoxicity and motor dysfunction [70]. Conversely, when administered in uncontrolled doses to dogs, ropivacaine demonstrated lower arrhythmogenicity than bupivacaine and lidocaine [69].

In the postoperative period, all animals received an NSAID (robenacoxib) until achieving the operated limb’s total weight bearing. Additionally, half of the animals were administered an adjuvant opioid drug (tramadol) for a duration of 15 days. The administration of the NSAID until TFWB of the limb aimed to provide analgesia and improve the tolerability of limb physical therapy. The addition of tramadol further enhanced the analgesic effect. It is worth noting that a previous study investigating the use of tramadol alone at a dose of 5 mg kg^−1^ for 10 days did not demonstrate any benefit in animals with elbow or stifle osteoarthritis [71].

The surgical procedure was performed on all animals by the first author (AK). Consistency was maintained across all cases regarding surgical technique and tissue manipulations. FHNE was performed using an osteotome and a mallet, as this technique is primarily utilized in our clinic, providing greater familiarity and expertise. After completing the ostectomy, the site was smoothed to eliminate any crepitus during passive hip movements. The anesthetist administered a local anesthetic or saline injection at the ostectomy site. The injection was allowed to remain for 5 min before the surgical field was closed. 

The procedure was performed in a blinded way by the surgeon.

The injection timing was selected to maximize the duration of the local anesthetic effect during the postoperative period and to prevent its removal during various manipulations, as tissue suturing was performed almost immediately. Lastly, intradermal suturing was employed to avoid the need for an Elizabeth collar, thereby minimizing postoperative anxiety in the animals.

All animals underwent physiotherapy and specifically flexion and extension movements immediately postoperatively, as recommended by many authors [72,73,74].

### 4.1. Weight Bearing of the Operated Limb

The assessment of postoperative progress in dogs with FHNE relied on evaluating the TIWB and TFWB of the limb. It is noteworthy and practical that a positive, strong, and statistically significant correlation was discovered between TIWB and TFWB in the operated limb. This observation implies that any of the studied factors leading to a decrease in TIWB will significantly shorten TFWB, thereby effectively enhancing the postoperative progression of FHNE. As indicated by the literature, early restoration of limb function is highly beneficial for achieving functional pseudarthrosis with a satisfactory range of motion [8].

Based on the results of the statistical analysis, only the first part of the study’s hypothesis appears to be valid. Specifically, the intraoperative administration of ropivacaine at the OS significantly contributed to a reduction in TIWB compared to the untreated animals, resulting in an earlier occurrence of initial weight bearing by 8.6 days on average. However, the additional use of epidural anesthesia preoperatively, either alone or in combination with postoperative tramadol administration, did not significantly reduce TIWB for animals receiving ropivacaine. The difference observed was only 1.4 and 0.4 days, respectively. It appears that the intraoperative administration of ropivacaine at the OS is sufficient to enhance the postoperative progression of FHNE, eliminating the need for adjunctive application of epidural anesthesia and/or postoperative tramadol administration.

An intriguing finding is a noteworthy decrease in TIWB resulting from the postoperative administration of tramadol (8.2 days), which was comparable to the equally significant reduction achieved by intraoperative ropivacaine administration (7 days). Consequently, in the context of FHNE, intraoperative ropivacaine, and postoperative tramadol administration appear to have a similar positive impact on FHNE’s postoperative progression. However, we propose the bolus administration of ropivacaine as a more practical technique.

The initial impression is that the addition of an opioid drug (tramadol) to enhance postoperative analgesia does not lead to a significant reduction in TFWB (70.4 days), thereby refuting the second hypothesis of the study. Similarly, the intraoperative administration of ropivacaine (66 days) has a comparable effect. Hence, it seems that tramadol and ropivacaine do not directly influence TFWB but rather indirectly affect it through the aforementioned positive correlation between the two time periods. However, upon examining the impact of preoperative epidural anesthesia in combination with intraoperative ropivacaine administration on TFWB, a significant reduction is observed (55 days), which further increases when tramadol is additionally administered postoperatively (33 days).

Lastly, when comparing the impact of intraoperative ropivacaine administration combined with preoperative epidural anesthesia and postoperative tramadol administration on TFWB to the individual effects of ropivacaine or tramadol on TIWB, it can be concluded that the utilization of all three analgesic techniques (multimodal analgesia) has the most significant effect in enhancing the postoperative progression of FHNE. This ultimately results in a shorter TFWB duration for the operated limb.

### 4.2. Pain

Based on the observations regarding the postoperative evolution of FHNE, particularly at the clinical level, following the implementation of different analgesic techniques, an endeavor was undertaken to elucidate their interpretation within the context of nerve pathways and pain.

Painful stimuli generated during surgical procedures lead to alterations in both nerve endings, triggering peripheral sensitization, and the spinal cord, inducing central sensitization. Primary and secondary hyperalgesia and allodynia are consequences of these changes [75]. While anesthesia eliminates the process of pain perception, it does not inhibit the transmission of impulses, which can lead to tachycardia, tachypnea, and increased blood pressure. Additionally, both peripheral and central sensitization, resulting from centrally transmitted impulses, contribute to heightened pain as the effects of anesthetic drugs wear off. Therefore, managing pre- and postoperative pain is pivotal in the overall clinical presentation, particularly when considering surgical wound inflammation and the subsequent persistent postoperative pain [76].

The Von Frey fibril principle, using an algometer, has been employed to evaluate postoperative pain following OHE in both cats [77,78,79] and dogs [80,81,82]. Similarly, this approach has been utilized in studies involving dogs with thoracolumbar muscle pain [83], chronic neuropathic pain [84], stifle and hip osteoarthritis [85], as well as after osteosynthesis [35] or stifle stabilization following cranial cruciate ligament rupture [86]. The algometer is a reliable means of assessing pain and gauging the level of analgesia by stimulating the nerve pathway responsible for transmitting painful stimuli (A and C nerve fibers), analogous to natural stimuli [87]. In accordance with previous canine studies, the chosen unit of measurement was grams [35,45,77,88].

It is worth noting that despite FHNE being a painful procedure, none of the animals required rescue analgesia and, as a result, were not excluded from the research protocol. It appears that the different combinations of perioperative analgesic drugs administered played a pivotal role in achieving this outcome. Although the control group exhibited significantly higher UMPS scores, they did not reach the threshold for necessitating rescue analgesia.

In addition to the algometer, the UMPS scale was employed to evaluate pain intensity. This scale was selected for its high precision, relying on criteria encompassing behavioral changes and various clinical parameters. Including multiple factors enhances its specificity and sensitivity while minimizing observer bias [89].

#### 4.2.1. Time-Standardized Areas under the Curve

Significant differences were observed only for the variables AUCstNOS and AUCstUMPS, while no significant difference was found for the variables AUCstOS and AUCstCHL. Group A and C exhibited a notable distinction in the AUCstNOS variable, primarily due to secondary hyperalgesia. This resulted in a greater reduction in the pain threshold around the lesion site in Group C [75,76,90]. The absence of a statistically significant difference between Groups A and C in the AUCstOS variable may potentially be attributed to the duration of ropivacaine’s action, as further elaborated below.

Significant statistical differences were observed in the variable AUCstUMPS between Groups A and B and A and C, indicating that dogs undergoing FHNE experience reduced pain with epidural anesthesia. Furthermore, the combination of epidural anesthesia and intraoperative ropivacaine contributes to an even greater reduction in pain intensity compared to the control group. These findings support the significant impact of combining epidural anesthesia and intraoperative ropivacaine on the limb’s TIWB (Table 6) in Groups A1–C1.

#### 4.2.2. Measurements of Mechanical Pain Threshold

Significant statistical differences in the mechanical pain pathway at the OS were observed between Groups A and C at various time points, namely t_0_, t_1_, t_2_, and t_4_, encompassing the postoperative period of up to 6 h (Figure 3). These differences can be attributed to the duration of ropivacaine’s action in diffusion techniques [68]. However, at time points t_5_ and t_6_, when the effect of ropivacaine had subsided, the observed differences were not statistically significant. This could explain the absence of a significant longitudinal difference in the variable AUCstOS between Groups A and C. It is possible that extending the duration of ropivacaine’s action could be beneficial, but the acidic environment resulting from inflammation in the surgical field may impede the effectiveness of local anesthetics, which are weak bases with a pKa of 7.7–9.1 [69,91,92]. Exploring alternative local anesthetics with lower pKa values than ropivacaine (pKa 8.1) could be an avenue for future research in this area.

As anticipated and demonstrated in Figure 3, Group A exhibited the highest pain threshold, while Group C displayed the lowest, with the most significant reduction observed in relation to the preoperative value.

In the “healthy” area NOS, a statistically significant difference was observed between Groups A and C at all postoperative time points, primarily attributed to secondary hyperalgesia. Figure 4 illustrates that, similar to the OS, Group A exhibited the highest pain threshold, while Group C demonstrated the lowest threshold.

In contrast to the aforementioned regions, the variations observed in the CHL were similar among all three groups (Figure 5), thus minimizing the influence of animal temperament in the study. Furthermore, it appears that the local anesthetic used had a localized rather than systemic effect, as evidenced by the consistent levels of measurements observed throughout.

#### 4.2.3. Measurements of Pain Intensity on the UMPS

According to the UMPS scale, the control dogs exhibited significantly higher pain intensity than the animals in Group A, indicating the beneficial analgesic effect resulting from the combination of epidural anesthesia with intraoperative administration of ropivacaine. Group C experienced the strongest intensity of pain among the groups, although rescue analgesia was not required. Conversely, Group A experienced the least pain intensity. In all groups, there was an initial increase in pain intensity between time points t_0_ and t_1_, which can be attributed to the partial recovery of the animals’ sensory function during the first postoperative measurement. Subsequently, pain intensity decreased over time.

## 5. Conclusions

In conclusion, implementing multimodal analgesia involving preoperative epidural anesthesia with morphine, intraoperative administration of ropivacaine at the OS, and postoperative administration of tramadol, combined with physiotherapy and NSAID, led to expedited weight bearing in dogs that have undergone FHNE.

## Figures and Tables

**Figure 1 animals-13-02295-f001:**
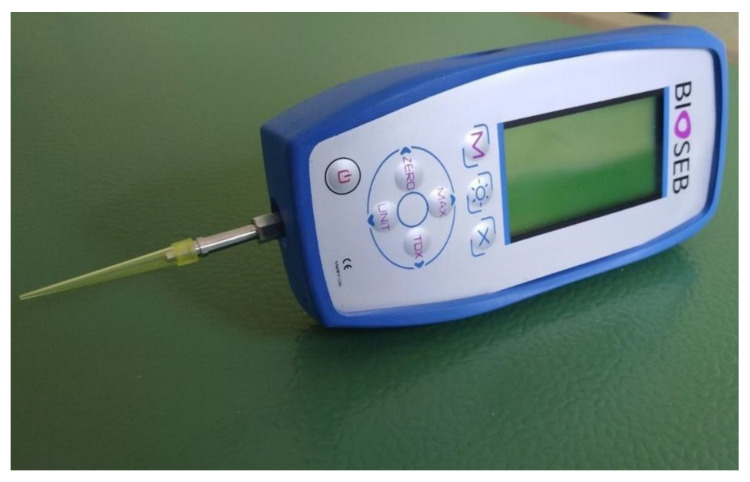
Von Frey filaments (algometer).

**Figure 2 animals-13-02295-f002:**
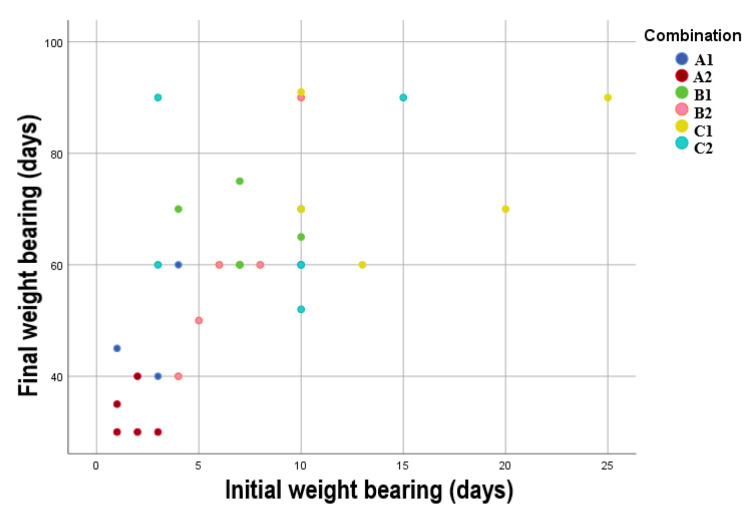
Scatterplot showing the correlation between initial and final limb weight bearing for all the subgroups.

**Figure 3 animals-13-02295-f003:**
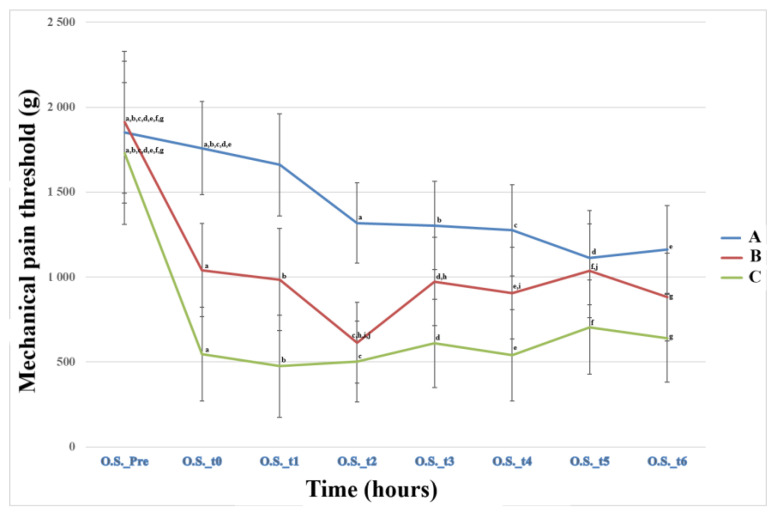
Means (±standard error) of mechanical pain threshold of the three groups at the ostectomy site (OS) of the operated limb related to time. Time (Pre): preoperative, (0) after the end of the operation and anesthesia, and also 1 (t_1_), 2 (t_2_), 4 (t_3_), 6 (t_4_), 20 (t_5_), and 24 h (t_6_) postoperative; data in the same row with the same letter in the superscription are significantly different from each other (*p* < 0.05).

**Figure 4 animals-13-02295-f004:**
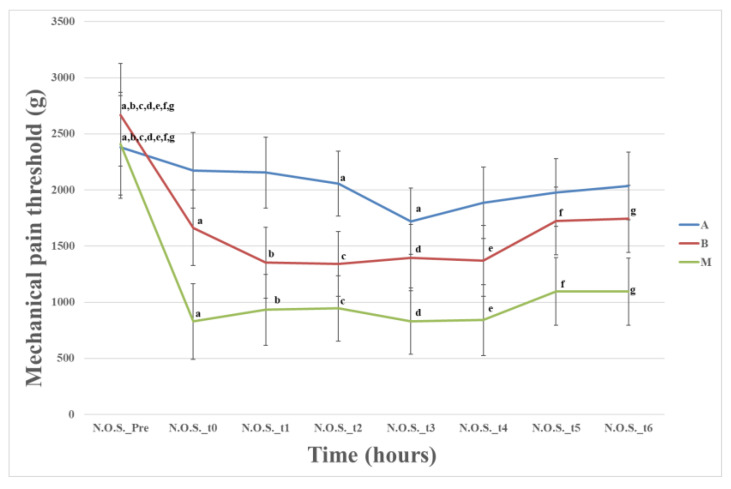
Means (±standard error) of mechanical pain threshold of the three groups at a “healthy” area near the ostectomy site of the operated limb related to time; Time (Pre): preoperative, (0) after the end of the operation and anesthesia, and also 1 (t_1_), 2 (t_2_), 4 (t_3_), 6 (t_4_), 20 (t_5_), and 24 h (t_6_) postoperative; data in the same row with the same letter in the superscription are significantly different from each other (*p* < 0.05).

**Figure 5 animals-13-02295-f005:**
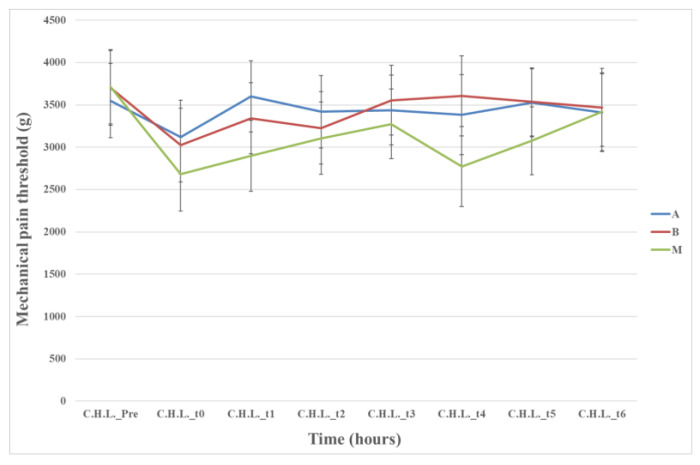
Means (±standard error) of mechanical pain threshold of the three groups of the contralateral healthy limb related to time; Time (Pre): preoperative, (0) after the end of the operation and anesthesia, and also 1 (t_1_), 2 (t_2_), 4 (t_3_), 6 (t_4_), 20 (t_5_), and 24 h (t_6_) postoperative.

**Figure 6 animals-13-02295-f006:**
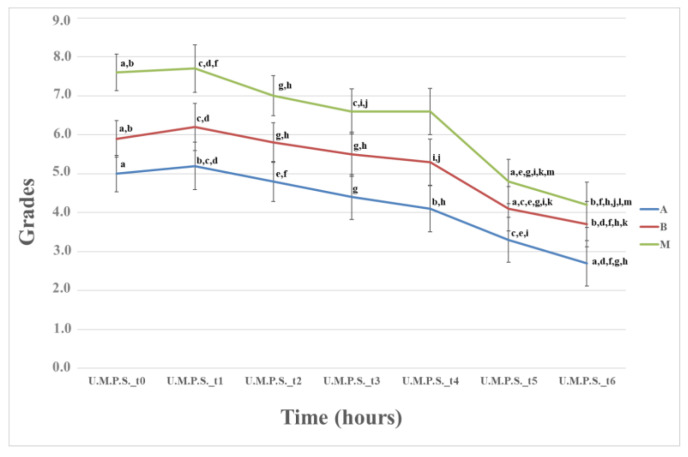
Means (±standard error) of grades of the University Melbourne Pain Scale of the three groups related to time; Time: (0) after the end of the operation and anesthesia, and also 1 (t_1_), 2 (t_2_), 4 (t_3_), 6 (t_4_), 20 (t_5_), and 24 h (t_6_) postoperative; data in the same row with the same letter in the superscription are significantly different from each other (*p* < 0.05).

**Table 1 animals-13-02295-t001:** Anesthetics and analgesics drugs administered in each group/subgroup.

	Groups/Subgroups					
	A (*n* = 10)		B (*n* = 10)		C (*n* = 10)	
	A1 (*n* = 5)	A2 (*n* = 5)	B1 (*n* = 5)	B2 (*n* = 5)	C1 (*n* = 5)	C2 (*n* = 5)
*Preoperative*	Dexdomitor (150 μg/m^2^, im) Tramadol (3 mg kg^−1^, iv) Robenacoxib (2 mg kg^−1^, sc) **Morphine (0.1 mg kg^−1^ epidural)**		Dexdomitor (150 μg/m^2^, im) Tramadol (3 mg kg^−1^, iv) Robenacoxib (2 mg kg^−1^, sc)		Dexdomitor (150 μg/m^2^, im) Tramadol (3 mg kg^−1^, iv) Robenacoxib (2 mg kg^−1^, sc)	
*Intraoperative*	**Ropivacaine (0.13 mL kg^−1^)**		**Ropivacaine (0.13 mL kg^−1^)**		Sodium chloride 0.9% (0.13 mL kg^−1^)	
*Postoperative*	Robenacoxib (1 mg kg^−1^ SID, p.o.)	Robenacoxib (1 mg kg^−1^ SID, p.o.) **Tramadol** **(3 mg kg^−1^ TID, p.o.)**	Robenacoxib (1 mg kg^−1^ SID, p.o.)	Robenacoxib (1 mg kg^−1^ SID, p.o.) **Tramadol** **(3 mg kg^−1^ TID, p.o.)**	Robenacoxib (1 mg kg^−1^ SID, p.o.)	Robenacoxib (1 mg kg^−1^ SID, p.o.) **Tramadol** **(3 mg kg^−1^ TID, p.o.)**

**Table 2 animals-13-02295-t002:** Lameness scale.

Degree of Lameness	Limb’s Weight Bearing				Characterization of Lameness
	Description	Stance	Walk	Run	
0	Full (normal) weight bearing	......	......	......	Absence
1	Partial weight bearing: hardly visible	......	......	......	Light
2	Partial weight bearing: easily visible	......	......	......	Mild
3	No weight bearing: intermittent, sporadic (≤1:5) *	......	......	......	Moderate
4	No weight bearing: intermittent, frequent (>1:5) *	......	......	......	Severe
5	No weight bearing: continuous	......	......	......	Not functional
Degree of lameness = (S + W + R)/3					

*: limb lift frequency per 5 steps.

**Table 3 animals-13-02295-t003:** Distribution of dogs’ breed submitted to femoral head and neck excision.

Breed	Number of Dogs
Mongrel	19
Labrador	3
Yorkshire terrier	2
Beagle	1
Chow-chow	1
Pitbull	1
Pomeranian	1
Poodle	1
Dogo Argentino	1
Total	30

**Table 4 animals-13-02295-t004:** Time of initial weight bearing of the operated limb (mean ± standard deviation).

	Time (days)	
	Subgroup 1	Subgroup 2
Group A	5.6 ± 4.159	1.8 ± 0.837
Group B	7.0 ± 2.121	6.6 ± 2.408
Group C	15.6 ± 6.656	8.2 ± 5.167

**Table 5 animals-13-02295-t005:** Comparison of subgroups in terms of time initial weight bearing of the limb subjected to femoral head and neck excision.

	A1	A2	B1	B2	C1	C2
**A1**		ns *p* = 0.153	ns *p* = 0.591	ns *p* = 0.701	* *p* = 0.001	ns *p* = 0.322
**A2**	ns *p* = 0.153		ns *p* = 0.055	ns *p* = 0.074	* *p* < 0.001	* *p* = 0.020
**B1**	ns *p* = 0.591	ns *p =* 0.055		ns *p* = 0.878	* *p* = 0.003	ns *p* = 0.645
**B2**	ns *p* = 0.591	ns *p =* 0.055	ns *p* = 0.878		* *p* = 0.003	ns *p* = 0.645
**C1**	* *p* = 0.001	* *p* < 0.001	* *p* = 0.003	* *p* = 0.002		* *p* = 0.008
**C2**	ns *p* = 0.322	* *p* = 0.020	ns *p* = 0.645	ns *p* = 0.540	* *p* = 0.008	

*: statistically significant, ns: not statistically significant.

**Table 6 animals-13-02295-t006:** Time of final weight bearing of the operated limb (mean ± standard deviation).

	Time (Days)	
	Subgroup 1	Subgroup 2
Group A	55.0 ± 12.247	33.0 ± 4.472
Group B	66.0 ± 6.519	60.0 ± 18.708
Group C	76.2 ± 13.682	70.4 ± 18.188

**Table 7 animals-13-02295-t007:** Comparison of subgroups in terms of the time of final weight bearing of the limb subjected to femoral head and neck excision.

	A1	A2	B1	B2	C1	C2
**A1**		* *p* = 0.016	ns *p* = 0.207	ns *p* = 0.561	* *p* = 0.020	ns *p* = 0.082
**A2**	* *p* = 0.016		* *p* = 0.001	* *p* = 0.004	* *p* < 0.001	* *p* < 0.001
**B1**	ns *p* = 0.207	* *p* = 0.001		ns *p* = 0.486	ns *p* = 0.241	ns *p* = 0.609
**B2**	ns *p* = 0.561	* *p* = 0.004	ns *p* = 0.486		ns *p* = 0.068	ns *p* = 0.232
**C1**	* *p* = 0.020	* *p* < 0.001	ns *p* = 0.241	ns *p* = 0.068		ns *p* = 0.501
**C2**	ns *p* = 0.082	* *p* < 0.001	ns *p* = 0.609	ns *p* = 0.232	ns *p* = 0.501	

*: statistically significant, ns: not statistically significant.

**Table 8 animals-13-02295-t008:** Means (±standard error) of preoperative measurements of mechanical pain threshold at the ostectomy site, in a “healthy” area near the ostectomy site, and in the contralateral healthy limb.

Mechanical Pain Threshold	Group A	Group B	Group C
Ostectomy site (g)	1852.2 ± 417.7	1911.3 ± 417.7	1727.8 ± 417.7
Healthy area near the ostectomy site (g)	2381.3 ± 457	2668 ± 457	2410.8 ± 457
Contralateral healthy limb (g)	3549.8 ± 439.4	3701.7 ± 439.4	3713.8 ± 439.4

## Data Availability

Not applicable.

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
