# Peer review of "Contribution to the Study of Perioperative Factors Affecting the Restoration of Dog’s Mobility after Femoral Head and Neck Excision: A Clinical Study in 30 Dogs"

_animals, 2023, doi:10.3390/ani13142295_

Round 1

Reviewer 1 Report

Contribution to the study of perioperative factors affecting the restoration of dog's mobility after femoral head and neck excision. A clinical study in 30 dogs

This is an interesting prospective clinical study designed to evaluate the role of perioperative analgesia during femoral head and neck ostectomy in dogs and the relationship of various analgesics protocols to the clinical recovery time of the operated limb

The study design is interesting and well set up

The keywords appear adequate; the term "multimodal analgesia" could be added to facilitate searching

The language used is clear easily understandable and the reading appears simple.

However, there are some observations that require a response or changes from the authors

173: Was the visual assessment of gait always done by the same person? I think it should be specified. In Table 2, the meaning of the values in the Stance, Walk and Run columns is not understood. Was the lameness scale designed by the authors or is it inspired by the literature?

205-218: the technical explanation of how the algometer works should not be done in materials and methods but in the introduction

651: "fracture": the correct description is ostectomy not fracture

677 - 678: this statement assumes that cases of licking or self-injury at the operative site are determined solely by skin suture. Unfortunately, I am not aware of this being the case. If the authors believe this to be so, they should clearly state it or cite appropriate bibliography

687-688: This statement seems to me to be a bit too assertive and deductive; it excludes the possibility that intercurrent factors, even those not considered in this study, could change the correlation between TIWB and TFWB observed by the authors in their case series

Do the authors not believe that the lack of use of a nonobjective system for assessing the weight load of the operated limb (force plate analysis) may be a limitation to the interpretation of the data (especially for intermediate lameness degrees)?

Do the authors believe that the number of cases used in the study is sufficient to reach reliable conclusions?

If the answer to these two questions is positive it should be reported in the conclusions just as if the authors believe there is reason for doubt these observations should be reported as a limitation of the study.

Author Response

We would like to thank the Reviewer and the Assistant Editor for the time they have spent to improve our work and for their comments. We have taken into consideration their suggestions in preparing the revised manuscript. Regarding your recommendations, we would like to provide the following:

  • This is an interesting prospective clinical study designed to evaluate the role of perioperative analgesia during femoral head and neck ostectomy in dogs and the relationship of various analgesics protocols to the clinical recovery time of the operated limb

            Thank you.

  • The study design is interesting and well set up

Thank you.

  • The keywords appear adequate; the term "multimodal analgesia" could be added to facilitate searching

It was added to the manuscript.

  • The language used is clear easily understandable and the reading appears simple.

Thank you. Because English is not our native language, the text was editing by specialized company before submission.

  • However, there are some observations that require a response or changes from the authors

Thank you for your observations.

  • Point 1. p. 173: Was the visual assessment of gait always done by the same person? I think it should be specified.

Response 1. Yes, it was always done by the first author. It was added to the manuscript.

  • Point 2. In Table 2, the meaning of the values in the Stance, Walk and Run columns is not understood. 

Response 2. In each column (stance, walk and run) the degree of lameness is noted, and the final grade is obtained by summing the three of them and then dividing by three (mean lameness degree).

  • Point 3. Was the lameness scale designed by the authors or is it inspired by the literature?

Response 3. It is inspired by the literature (1. Hazelwinkel HAW, Meij BP, Theyse LFH, van Rijssen B. Locomotor System In Medical History and Physical Examination in Companion Animals Saunders Elsevier 1990 p.139

2. Vasseur PB and Slatter D. Musculoskeletal System. In Textbook of Small animal Surgery 2nd ed.; Slatter, D., Ed.; WB Saunders: Philadelphia, USA, 1993; p.1578

  1. Brunnberg L. Visual Examination of the animal in Motion. In Diagnosing Lameness in Dogs Johnson, K.A., Ed.; Blackwell Science: Oxford, UK, 1998; p. 28), but the final combination/design was done by Prof N. Prassinos (co-author). This lameness scale is used in our Clinic at least for the last 15 years, we think that it is reliable and that is the reason we chose this one for our study.
  • Point 4. pp. 205-218: the technical explanation of how the algometer works should not be done in materials and methods but in the introduction. 

Response 4. Although we prefer the technical details about algometer to be presented in the Materials and Methods section, they have been moved in the Introduction section.

  • Point 5. p. 651: "fracture": the correct description is ostectomy not fracture. 

Response 5. The word “fracture” refers to reference No 35, not to our study. In that reference, bupivacaine was injected directly into the fracture site of long bones’ diaphysis.

  • Point 6. 677 - 678: this statement assumes that cases of licking or self-injury at the operative site are determined solely by skin suture. Unfortunately, I am not aware of this being the case. If the authors believe this to be so, they should clearly state it or cite appropriate bibliography.

Response 6. The authors do not assume that cases of licking or self-injury at the operative site are determined solely by skin suture. It is well known that the complications of a skin suture are  problems in aesthetic appearance, scar development, and others, such as infection, rupture, granulation, necrosis, serous collection, and hematoma. The use of an Elizabeth collar protects the suture line only from the complications originated from the animals’ licking or self-injury, as suture removal, while on the other hand causes animals’ anxiety. Also, it is well known than intradermal suturing itself protects suture removal by the animal. So, in an attempt to minimize the possibilities of suture removal by the dogs and to eliminate the animals’ anxiety, we decided to use intradermal suture and no Elizabeth collar.

  • Point 7. pp. 687-688: This statement seems to me to be a bit too assertive and deductive; it excludes the possibility that intercurrent factors, even those not considered in this study, could change the correlation between TIWB and TFWB observed by the authors in their case series.

Response 7. Although the above statement originates from the statistical analysis of our data, it is wrong to generalize the conclusion. Of course, the correlation between ΤΙΒ and TFWB refers only to the studied parameters. For another, maybe unknown, factor the correlation could be different. So, the lines 687-688 were modified accordingly: “This observation implies that any of the studied factors leading to a decrease in TIWB will significantly shorten TFWB, thereby effectively enhancing the postoperative progression of FHNE”.

  • Point 8. Do the authors not believe that the lack of use of a nonobjective system for assessing the weight load of the operated limb (force plate analysis) may be a limitation to the interpretation of the data (especially for intermediate lameness degrees)?

Response 8. We are not sure that the term “limitation” describes accurately the lack of force plate analysis. Of course, the latter provides more objective results, but in our study, we reinforced the visual observation of the patient’s gait, with the thoughtful study of the video and slow motion, obtained when the locomotion of the dogs was examined.

  • Point 9. Do the authors believe that the number of cases used in the study is sufficient to reach reliable conclusions?

Response 9. A statistician had advised us throughout the study. Therefore, at the beginning, a power analysis was conducted to ensure an ethically acceptable study while minimizing the number of animals required to achieve the scientific objectives.

Reviewer 2 Report

Reviewer comments on “animals-2492331” manuscript

Please find specific in-line comments in the attached pdf file; here are some summarizing and general comments:

This prospective study has several flaws and limitations which should be addressed, corrected and where not possible at least discussed and detailed in a separate section (“limitations of the study”) at the end of the Discussion and before the Conclusions. 

To investigate and propose improved post-op pain control in debilitating and invasive surgeries such as FHNE is important, is of interest, and warrants publication. Overall, the study design is appropriate, yet the division of an already modest cohort of 30 cases into 6 groups (3 main and 3 sub-groups) reduces the power of the results and renders evidence questionable; this, despite sophisticated statistical tools including a positive power analysis. However, mathematics do not take into account the large (and uncontrollable) number of variables and possible confounding factors: 1) each dog reacts differently to pain – either the post-operative pain and/or to painful stimuli. Of course, this problem exists in any clinical (orthopaedic) study, yet to overcome this variable, groups larger than 5 dogs/group may be necessary, despite considering statistically a “within” subject factor ; 2) no FHNE procedure has an identical (and sometimes not even comparable) outcome despite one surgeon’s hand; 3) the underlying diseases/lesions as indications for FHNE in your cohort included 5 different pathologies, each inheriting side effects, local and/or systemic, difficult to evaluate, but most likely effectuating different recovery times, linked not only to the surgical procedure and the analgesic protocol, but also to the underlying pathology;  4) physical therapy, although an important rehabilitating factor, adds another bias to the study, as to physiotherapy, active or passive, each individuum reacts differently; 5) the point in time when the animal first began putting weight on the operated limb (FIWB) is a difficult to assess moment – some dogs initiate “tipping”, then for a time (even days)  lift the leg until they initiate bearing weight again….    

Having said this, this reviewer of course recognizes that a clinical prospective study cannot be ideal and that confounding factors may be unavoidable. Yet, please mention them, analyze them and discuss them in “limitations”. When undergoing a Major Revision, I believe your paper will find strong consideration for publication, also because the conclusions you draw are appropriate as they stand (without “overshooting”): “multimodal analgesia pre-,intra-, and post-op combined with physiotherapy and early return to function are the keys for good long-term outcome.” 

PS. Epidural anaesthesia is an intra-operative procedure (although obviously applied in the preparation phase) 

Author Response

We would like to thank the Reviewer and the Assistant Editor for the time they have spent to improve our work and for their comments. We have taken into consideration their suggestions in preparing the revised manuscript. Regarding your recommendations, we would like to provide the following:

  • This prospective study has several flaws and limitations which should be addressed, corrected and where not possible at least discussed and detailed in a separate section (“limitations of the study”) at the end of the Discussion and before the Conclusions. 

We appreciate your effort to improve our manuscript. We will try to ameliorate it according to your comments.

  • To investigate and propose improved post-op pain control in debilitating and invasive surgeries such as FHNE is important, is of interest, and warrants publication.

Thank you.

  • Overall, the study design is appropriate, yet the division of an already modest cohort of 30 cases into 6 groups (3 main and 3 sub-groups) reduces the power of the results and renders evidence questionable; this, despite sophisticated statistical tools including a positive power analysis. However, mathematics do not take into account the large (and uncontrollable) number of variables and possible confounding factors

Although in our study, the clinical meaning of the results is obvious, even without the use of statistics, the decision of a journal referee would be negative if the results are not supported statistically. This is why a statistician is needed throughout the study to conduct the researchers. You are right that sometimes mathematics does not consider the large (and uncontrollable) number of variables and possible confounding factors, but if the results are based on statistical processing?  In our study, the number of dogs was conducted to ensure an ethically acceptable study, while minimizing the number of animals required to achieve the scientific objectives.

  • each dog reacts differently to pain – either the post-operative pain and/or to painful stimuli. Of course, this problem exists in any clinical (orthopaedic) study, yet to overcome this variable, groups larger than 5 dogs/group may be necessary, despite considering statistically a “within” subject factor.

Response. Please, believe us, we would like to study groups of multiple animals than the ones in our study. However, this was not possible. We were also unlucky to meet two quarantines due to COVID-19 during the study period, but generally it seems that is an elaborate work based on rightful rules, as mentioned in the previous question.

  • 3) the underlying diseases/lesions as indications for FHNE in your cohort included 5 different pathologies, each inheriting side effects, local and/or systemic, difficult to evaluate, but most likely effectuating different recovery times, linked not only to the surgical procedure and the analgesic protocol, but also to the underlying pathology. 

Response. Thank you for the question. It would be ideal to have groups of 5 or more animals for each pathology. However, we believe that these diseases could be divided into acute and chronic ones and the most important parameter in their evaluation is the presence of muscle atrophy at the time of FHNE. We were very thoughtful in the selection of the cases included in this study, none of which had severe muscle atrophy even in the diseased limb.

  • 4) physical therapy, although an important rehabilitating factor, adds another bias to the study, as to physiotherapy, active or passive, each individuum reacts differently.

Response. In our study, we tried to apply the standard management/treatment in all cases, as it is proposed in the classic orthopaedic books. Physiotherapy is part of this treatment. In our opinion, physiotherapy, even in its simplest form, helps the patients essentially to recover from FHNE and other orthopaedic surgeries. The different reaction of each individual mainly depended on the pain sensation. This is another positive action of effective postoperative analgesia.

  • 5) the point in time when the animal first began putting weight on the operated limb (FIWB) is a difficult to assess moment – some dogs initiate “tipping”, then for a time (even days)  lift the leg until they initiate bearing weight again….  

Response. You are right about this. It can happen, but this did not happen to any animal of the current study. The follow up was very strict.

  • also because the conclusions you draw are appropriate as they stand (without “overshooting”): “multimodal analgesia pre-,intra-, and post-op combined with physiotherapy and early return to function are the keys for good long-term outcome.” 

Response. We added it.

  • Epidural anaesthesia is an intra-operative procedure (although obviously applied in the preparation phase) 

Response. We say that it is a preoperative procedure. Of course, it has an intraoperative effect such as antibiotics or NSAIDs which are also administrated preoperative.

  • How long is a postoperative period?

Response. In this study (ref  No22) in humans, the analysis of the records of 929 patients who had undergone orthopaedic surgery, shows the median time to first request for postoperative analgesia following different forms of therapy. So, the longest period was more than 9 hours after opiate premedication and local anaesthesia.

  • in addition to that an institutional ethics committee approval is necessary for publication - please provide protocol number

Response. It was written at the end in: Institutional Review Board Statement. We added it also at the beginning of materials and methods.

  • what is the rational applied to the choice of 4 months ? please explain.

Response. According to the SPC of Onsior, which had to be taken for a long period of time, the safety of the veterinary medicinal product has not been established in dogs less than 3 months of age. Based on this we decided to choose animals over 4 months to ensure greater safety.

  • suggest highlighting the differences between groups either with colour or bold print asterixes or alike.

Response. Ok

  • since pain assessments using this device are very infrequently used in veterinary medicine, besides this (unnecessary ? image) I suggest describing briefly what it entails (filament which bend and filament diameter increasing until pain reaction and this expressed in grammes of filament pressure on the skin as pain threshold)

Response. As you state, algometer is very infrequently used in veterinary medicine, so we consider that its photo is useful for the reader to understand its function. We also added some information about its operation, which help anyone to use it for research and clinical purposes. 

  • what is the purpose of expressing breed distriburion in % ? and, in any case, there were only 30 dogs.

Response. Because this manuscript is part of a PhD Thesis, some data are presented in detail. The column was deleted.

  • the enormous variation in underlying diseases and their pre-op duration are of major concern for post-op and long-term pain evaluation; muscle atrophy and many other factors such as tolerance to nociceptive stimuli developing over time are only some possible confounding factors.

Response. Thank you for the comment. You touched on two crucial interactive factors in the study of this surgical procedure. Certainly, the factor “underlying disease” has a wide variety, but these diseases could divide into two categories: acute and chronic. Unlike acute diseases, the chronic ones could obviously affect muscle mass, which is an important parameter in the post-op progress of FHNE. However, the real problem is when the decision for the FHNE procedure is takenË™ before or after severe muscle atrophy has been established. The other factor “pre-op duration” has also a wide range for many causes (e.g., chronic diseases, lameness degree, owner decision, stray dogs). However, the question is the same as muscle mass is also the important factor for the post-op FHNE progressË™ was there muscle atrophy pre-op? It is our omission that in the animals’ criteria we did not include that dogs with severe muscle atrophy were excluded from the study (could we add this criterion now?). Indeed, we were very meticulous in the selection of cases and some dogs with muscle atrophy and uncertain history were not included. Although we used dogs with chronic diseases (range 5-420 days ), you can see that the mean (61 days) and the median (28 days) pre-op duration were relatively short, but the most important was that the affected limb muscle mass of the few animals with a chronicity of more than 3 months was satisfactory.

  • this repeated explanation that there was no significant difference when you say in the same context there was no significant effect is unnecessary and confusing; please shorten and adapt this throughout the text.

Response. In the Results it is necessary to emphasize the statistical significance of our data, while we understand that it is tiring. We are sure that you appreciate that it is not continued in the Discussion!

  • Lines 580-585

Response: We have shortened them.

  • you might not like this comment and you do not have to reflect it in your paper but when you talk about technical improvements of the excision technique, I just like to mention that hip joint replacenet is a modern, efficient and well described alternative (even in very small dogs and now also in cats); of course, the results of yout study are well applicable to prosthetic implant surgery of the hip.

Response. Of course, hip joint replacement is a modern, efficient and well described alternative, but as we said in our introduction, FHNE is a cheaper and also reliable procedure. It is a good idea for a survey that investigates how often each operation is done all over the world. 

  • perfect, but unfortunately an additional potential bias

Response. In our opinion, physiotherapy, even in its simplest form, helps the patients essentially to recover from FHNE and other orthopaedic surgeries. The classic orthopaedic books also refer to this statement.

  • What progression?

Response. FNHE’s postoperative progression

  • why pre-operative ? of course it is applied during the pr-op prep (like endotracheal intubation), but it is an intra-operative analgesic.

Response. We write that it is a preoperative procedure. Of course, it has an intraoperative effect, such as antibiotics or NSAIDs which are also administrated preoperative.

All the other corrections have been incorporated into the manuscript.

Round 2

Reviewer 1 Report

I have nothing further to suggest, in its current form the article appears suitable for publication

Reviewer 2 Report

thank you for your thoughtful replies, comments and textual revisions; recommend acceptance for publication